# FTY720-P, a Biased S1PR Ligand, Increases Mitochondrial Function through STAT3 Activation in Cardiac Cells

**DOI:** 10.3390/ijms24087374

**Published:** 2023-04-17

**Authors:** Juan Pablo Muñoz, Paula Sànchez-Fernàndez-de-Landa, Elena María Goretti Diarte-Añazco, Antonio Zorzano, Francisco Blanco-Vaca, Josep Julve

**Affiliations:** 1Institut d’Investigació Biomèdica Sant Pau (IIB SANT PAU), 08041 Barcelona, Spain; 2CIBER de Diabetes y Enfermedades Metabólicas Asociadas (CIBERDEM), 28029 Madrid, Spain; 3Institute for Research in Biomedicine (IRB Barcelona), 08028 Barcelona, Spain; 4Departament de Bioquímica i Biomedicina Molecular, Facultat de Biologia, Universitat de Barcelona, 08028 Barcelona, Spain; 5Department of Clinical Biochemistry, Hospital de la Santa Creu i Sant Pau, 08041 Barcelona, Spain; 6Department de Bioquímica i Biologia Molecular, Universitat Autònoma de Barcelona, 08041 Barcelona, Spain; 7Department of Endocrinology and Nutrition, Hospital de la Santa Creu i Sant Pau, 08041 Barcelona, Spain

**Keywords:** FTY720-P, mitochondria, TFAM, STAT3, DRP1 S616, cardiomyocytes, metabolism, S1PR, ATP, nucleoids

## Abstract

FTY720 is an FDA-approved sphingosine derivative drug for the treatment of multiple sclerosis. This compound blocks lymphocyte egress from lymphoid organs and autoimmunity through sphingosine 1-phosphate (S1P) receptor blockage. Drug repurposing of FTY720 has revealed improvements in glucose metabolism and metabolic diseases. Studies also demonstrate that preconditioning with this compound preserves the ATP levels during cardiac ischemia in rats. The molecular mechanisms by which FTY720 promotes metabolism are not well understood. Here, we demonstrate that nanomolar concentrations of the phosphorylated form of FTY720 (FTY720-P), the active ligand of S1P receptor (S1PR), activates mitochondrial respiration and the mitochondrial ATP production rate in AC16 human cardiomyocyte cells. Additionally, FTY720-P increases the number of mitochondrial nucleoids, promotes mitochondrial morphology alterations, and induces activation of STAT3, a transcription factor that promotes mitochondrial function. Notably, the effect of FTY720-P on mitochondrial function was suppressed in the presence of a STAT3 inhibitor. In summary, our results suggest that FTY720 promotes the activation of mitochondrial function, in part, through a STAT3 action.

## 1. Introduction

FTY720 is an immunosuppressant currently used for the treatment of relapsing–remitting multiple sclerosis [1]. The phosphorylated form of this compound interacts with sphingosine-1-phosphate (S1P) receptor 1, S1PR1, localized in lymphocytes, and promotes its internalization and degradation. In this way, suppression of the S1P signal blocks the egress of lymphocytes from secondary lymphoid organs to the blood, leads to lymphopenia, and decreases the inflammatory response [2]. In the context of cardiovascular diseases, in vivo administration of FTY720 improves cardiac function in heterotopic heart transplantation models and in hearts of diabetic rats [3,4]. Interestingly, cardiac amelioration was accompanied by reduced oxidative stress and apoptosis and repressed cardiac fibrosis. Supporting these findings, the administration of FTY720 also improved cardiac dysfunction and decreased fibrosis in recent independent studies using animal models of genetically induced hypertrophic cardiomyopathy and in myocardial ischemic/reperfusion [5,6,7]. The administration of FTY720 favorably influences glucose sensitivity in non-obese diabetic mice (NOD), in db/db mice, and in a non-human primate model [8,9,10]. Moreover, it promotes lipolysis upon consumption of a high-fat diet (HFD) [11]. Some studies propose that the anti-inflammatory and antioxidant effects of FTY720 improve energy metabolism. However, the effects of phosphorylated FTY720 (FTY720-P), the active ligand of S1PRs [12], on mitochondrial functionality is poorly explored.

Mitochondrial metabolism is required for ATP production, Ca^2+^ homeostasis, lipid synthesis, and amino acid production in cardiac cells [13,14,15]. Mitochondrial dysfunction in vivo is linked to insulin resistance, muscle atrophy, fibrosis, and liver inflammation and leads to cardiac disease [16]. Thus, finding new compounds that promote mitochondrial function could enable novel treatments of metabolic diseases. Currently, repurposing FDA-approved drugs is an emergent and promising field that generates great advances in metabolic disease treatment [17].

Mitochondrial homeostasis is sustained by mitochondrial dynamics, mitochondrial biogenesis, and mitochondrial turnover processes. Dynamin-related protein 1 (DRP1) is a GTPase required for mitochondrial fission. This protein is phosphorylated at serine 616 by ERK1/2, CDK1, or CAMKII and is recruited to the mitochondrial outer membrane [18,19,20]. Moreover, recent studies suggest that mitochondrial fission is related to mitochondrial DNA (mtDNA) replication and segregation [21,22]. The mitochondrial genome encodes 37 genes including tRNA, rRNA, and 13 subunits of the OXPHOS complex I, III, IV, and V [23]. Mutations or reduced content of mtDNA are frequently associated with several metabolic diseases [24]. Indeed, alterations in the expression of proteins, such as mitochondrial transcription factor A (TFAM), that are involved in mtDNA replication or transcription generally lead to mitochondrial dysfunction. This protein is finely regulated and promotes the activation in mtDNA replication and transcription [25]. Conversely, several studies have demonstrated that an increase in mtDNA replication improves muscle function in human patients upon exercise [26].

Mitochondrial biogenesis is a coordinated process that requires the activation of expression of mtDNA and nuclear-encoded mitochondrial genes to generate mitochondrial components, such as proteins, phospholipids, and cofactors [27]. The co-activator PGC1-alpha interacts with nuclear respiratory factor 1 (NRF1), ERRα, and PPARs and promotes the expression of several genes required for mitochondrial biogenesis, including OXPHOS genes, mitochondrial protein translocation genes, and TFAM [27]. In addition, other transcription factors have been involved in mitochondrial biogenesis, such as c-Myc and STAT3 [28,29].

Signal transducer and activator of transcription (STAT3) is a transcription factor that modulates a plethora of cellular process and mediates the intracellular signaling of several cytokines [30]. STAT3 transcriptional activity is triggered by phosphorylation at tyrosine 705 by the JAK pathway [31]. Beyond inflammation, STAT3 also influences mitochondrial homeostasis. For instance, STAT3 ablation in β-pancreatic cells impairs glucose metabolism and reduces mitochondrial genes expression and mitochondrial function. In addition, STAT3 expression promotes mitochondrial biogenesis [28,32] and mitochondrial respiration during myogenic linage progression through activation of the expression of FAM3a [33], a cytokine-like protein that modulates muscle development through paracrine and/or autocrine. Overall, these studies indicate a close relation between mitochondrial function and STAT3 transcriptional activity.

Recent evidence suggests that STAT3 signaling also modulates S1P signaling. First, JAK2-STAT3 signal inhibitor decreases the FTY720-dependent protective effects in white matter (WM) ischemic injury and acetaminophen-induced liver injury mouse models [34,35]. In contrast, FTY720 suppresses STAT3 activation in a chronic colitis disease model and B-cell lymphoma cells lines [36,37]. Although the basis of such distinct results is unknown, it might be related to tissue-specific patterns or to the FTY720 concentration used in different studies. Noteworthy, recent data evidenced that FTY720-P administration can sustain ATP and creatine levels in hearts, reduce mitochondrial ROS production in menadione-treated neurons, and increase the mitochondrial mass of bone-marrow-derived dendritic cells when co-incubated with LPS [38,39,40].

In this report, we demonstrate for the first time that FTY720-P promoted favorable structural and functional changes in mitochondrial physiology in a cardiac human cell line AC16. Furthermore, we show that STAT3 signaling is required for FTY720-P-dependent mitochondrial activation.

## 2. Results

### 2.1. FTY720-P Increases Mitochondrial Respiration

Nanomolar blood FTY720-P concentrations provide protection against inflammation and heart disease in mice [38]. In addition, FTY720-P improves glucose metabolism in mice made obese with a high-fat diet [41]. Based on this information, we explored the potential effect of FTY720-P treatment on the mitochondrial metabolism of the human cardiomyocytes cell line AC16. Time course studies of oxygen consumption rate (OCR) using a cell-live, real-time Seahorse respirometer show an increase in basal mitochondrial respiration after 48 h in the presence of 100 nM FTY720-P (Appendix A). FTY720-P-treated cells exhibited an increase in the OCR profile evaluated in the presence of an ATPase inhibitor (oligomycin), a mitochondrial proton ionophore (CCCP), and complex I and III OXPHOS inhibitors (Figure 1a). Moreover, FTY720-P-pretreated AC16 cells showed a significant increase (1.8-fold, *p*-value < 0.05) in basal and ATP-linked respiration (Figure 1b,c). In contrast, the extracellular acidification rate (ECAR) was not altered in these cells (Figure 1d). Our data demonstrated that FTY720-P specifically increased the ATP production rate and OXPHOS metabolism in AC16 cells (Figure 1e,f). We further analyzed whether the metabolic oxidation of energy substrates was influenced by the pretreatment of cells with FTY720-P. CO_2_ production was slightly but not significantly increased in cells incubated with radiolabeled ^14^C glucose upon FTY720-P treatment under our working conditions (Appendix A). Furthermore, no significant changes were observed in mitochondrial membrane potential (Figure 1h). Some investigations have demonstrated that a reduction in mitochondrial superoxide levels was linked with an improvement in mitochondrial bioenergetics parameters [42]. In this regard, we observed a significant decrease in mitochondrial superoxide levels in AC16 cells treated with FTY720-P compared with untreated cells (Figure 1i). This finding is consistent with previous evidence showing FTY720-mediated protection against oxidative stress [39]. Thus, our data support the concept that FTY720-P promotes mitochondrial function and decreases mitochondrial superoxide levels.

### 2.2. FTY720-P Alters Mitochondrial Morphology

Mitochondrial function is linked to morphological changes in the mitochondrial network [43]. We performed mitochondrial morphology analysis per cell using the ImageJ plugin Mitochondrial Analyzer. This provides information on the mitochondrial shape and mitochondrial network complexity. We observed a significant increase in mitochondrial number and mitochondrial sphericity upon FTY720-P treatments. Despite no alteration in total mitochondrial volume or mitochondrial branch number, we observed a significant increase in the mitochondrial branched endpoint. These data suggest that FTY720-P treatment induced the reorganization of the mitochondrial network in these cells. Mitochondrial dynamics are mediated by a coordinated action between fusion and fission proteins [44]. Immunodetection of mitochondrial dynamic proteins showed an increase in DRP1 phosphorylation at serine 616, a critical amino acid residue involved in DRP1 activation and mitochondrial fission [18,45] (Figure 2i,j). Conversely, the relative levels of the mitochondrial fusion protein, optic atrophy 1 (OPA1), did not differ between treated and non-treated cells (Figure 2i,j). Taken together, these results suggest that FTY720-P alters mitochondrial morphology at least in part by activating DRP1.

### 2.3. FTY720-P Increases the Expression of Proteins Involved in Mitochondrial Biogenesis

The enhanced activity of mitochondrial function observed in FTY720-P-treated AC16 cells could be associated with an increase in their mitochondrial mass; thus, we explored the effect of FTY720-P on the expression of TFAM and Translocase of Outer Mitochondrial Membrane 20 (TOM20) in AC16 cells. The relative protein abundance of both TFAM and TOM20 was increased in total cell protein extracts obtained from FTY720-P-treated cells. Moreover, their relative content was significantly increased in mitochondrial-enriched cell extracts (Figure 3a–d). Interestingly, protein changes were not linked to changes in mitochondrial membrane staining, as assessed by mitochondrial staining with Mitotracker Deep Red (Figure 3e). In addition, the relative abundance of the co-activator PGC1-alpha and the nuclear respiratory factor 1 (NRF1) were not altered in nuclear fractions under these conditions (Figure 3f,g). 

### 2.4. FTY720-P Increases the Number of Nucleoids

TFAM plays a crucial role in maintaining the mtDNA levels, and its expression and distribution are finely regulated [25]. Based on the increased expression of TFAM in FTY720-P-treated cells, we analyzed the number and size of mitochondrial nucleoids using super-resolution illumination microscopy (SIM) (Figure 4a–d). The mitochondrial nucleoid number was significantly elevated upon FTY720-P treatment (Figure 4c). Furthermore, the size of mitochondrial nucleoids was reduced in FTY720-P-treated cells (Figure 4d). In line with the decrease in the nucleoid size upon FTY720-P treatments, a recent report demonstrates that mutations and depletion of DRP1 lead to the formation of larger mitochondrial nucleoids [21]. Our data would be consistent with the view that DRP1 activation can modulate nucleoid size [21,46] (Figure 2c,d).

### 2.5. STAT3 Signal Was Required to Promote the FTY720-P-Dependent Mitochondrial Metabolism

Several reports have demonstrated that FTY720-P reduces STAT3-P (T705) and represses the inflammatory response in pathological mouse models [34,35]. Remarkably, FTY720-P induced the phosphorylation of JAK2 (Y1007/1008) and STAT3 (Y705) (pSTAT3) (Figure 5a,b). To obtain further insight on STAT3 activation, we further evaluated the subcellular distribution of pSTAT3 (Y705) in AC16 cells treated with FTY720-P. Our results showed increased immunostaining for pSTAT3 in the cellular membrane and cluster formation upon FTY720-P treatment (Figure 5c,d), suggesting that FTY720-P at nanomolar concentrations could be driving chronic activation of pSTAT3 (pY705). 

Because STAT3 activation has been linked to increased mitochondrial biogenesis [32], we explored whether mitochondrial function was suppressed by the specific STAT3 inhibitor S3I-201. This inhibitor binds the STAT3-SH2 domain and blocks its transcriptional activity [47]. Consistently, the relative abundance of TFAM was decreased in FTY720-P-treated cells in the presence of STAT3 inhibitor S3I-201 (Figure 6a,b). In parallel, the mitochondrial basal respiration rate was decreased in AC16 cells treated with FTY720-P and in the presence of the STAT3 inhibitor, as revealed by the reduction in the ATP-linked respiration rate, and maximal mitochondrial capacity in these conditions (Figure 6c–f). No significant changes were observed in either mitochondrial leak or ECAR under these conditions (Figure 6g,h). The presence of STAT3 inhibitor reduced the mitochondrial ATP production rate in FTY720-P-treated cells, without changes in the glycolytic ATP production rate (Figure 6i,j). The energy profile, as a surrogate of OXPHOS activity, was concomitantly decreased by the STAT3 inhibitor (Figure 6k). Overall, our data document that STAT3 promotes FTY720-P-dependent mitochondrial activation in AC16 cells.

## 3. Discussion

FTY720-P is a S1PR ligand that decreases inflammation and promotes survival and metabolic activation [12,38,41,48]. This compound induces the activation of several pro-survival signals, associated with a protective effect in tissues such as the brain and heart [7,49]. FTY720-P decreases the anti-inflammatory response by blocking the S1P signaling in T lymphocytes. This compound promotes S1PRs internalization and prevents lymphocytes from egressing from lymphoid tissues to the blood [2]. In addition, some other reports have also evidenced that FTY720-P can decrease prostaglandin E_2_ production in IL1β-treated mesangial and lung epithelial cells [50,51]. 

FTY720 improves glucose homeostasis in non-obese diabetic (NOD) mice, in db/db mice, and a in non-human primate model of spontaneous diabetes (NHP) [8,9,10]. Moreover, this compound promotes lipolysis activation and prevents obesity in a high-fat diet (HFD) mouse model [11]. These studies give rise to the question of whether FTY720-P could be directly modulating cellular metabolism. Thus, our data provide, for the first time, direct evidence of a FTY720-P-dependent activation of mitochondrial function and biogenesis in AC16 cardiac cells (Figure 1a–g). Of note, our study was carried out using nanomolar therapeutic concentrations of FTY720-P, which are within the therapeutic range. Some investigations indicate that micromolar FTY720 concentrations induce mitochondrial dysfunction and apoptosis in some cancer cell models [52].

The effect of FTY720-P on mitochondrial function is poorly defined; however, some evidence suggests that it may favorably influence mitochondrial homeostasis. First, FTY720-P treatment decreases mitochondrial ROS production upon menadione treatment in neuronal cells [39]. Another report shows that bone-marrow-derived dendritic cells co-incubated with LPS and FTY720-P increase mitochondrial mass as compared with LPS-treated cells [40]. Moreover, FTY720-P can stabilize ATP and creatine levels in a heterotopic model of heart transplantation [38]. Altogether, these data suggest a close link between FTY720-P signaling and mitochondrial function. In this regard, our results indicate that FTY720-P promotes mitochondrial respiration without an effect on aerobic glycolysis. In addition, we also observed a significant decrease in mitochondrial superoxide generation upon FTY720-P treatment (Figure 1h), being the latter in agreement with the antioxidant effect of FTY720 observed in the heart, brain, and liver [7,35,53]. 

Intriguingly, we observed a slight, but not significant, increase in glucose oxidation under our experimental conditions (Appendix A). Our data were in contrast with other research reporting that FTY720-P can activate glucose uptake in soleus muscle of HFD mice ex vivo [41]. It could be postulated that FTY720-P could also be promoting the oxidation of other metabolic substrates, such as pyruvate or amino acids, but this requires further investigation.

We observed an increase in the mitochondrial number and mitochondrial branches endpoint in cells treated with FTY720-P (Figure 2a–h). These observations suggest that FTY720-P treatment modified mitochondrial architecture. Further, studies using time-lapse cell live microscopy may shed light on mitochondrial dynamics in these cells.

Such mitochondrial network modification, accompanied by an increase in the relative abundance of phosphorylated DRP1 (serine 616), is associated with an activation of mitochondrial dynamics and metabolism (Figure 2i,j). Consistently, DRP1 activation promotes mitochondrial fission and metabolism activation and its expression ablation leads to alterations in mitochondrial respiration [45,54,55].

Fine-tuning regulation of TFAM-to-mtDNA ratio is crucial to sustain a correct mtDNA replication and mitochondrial function [25]. Our results suggest an activation of mitochondrial biogenesis in FTY720-P-treated cells (Figure 3a–d). However, mitochondrial mass, assessed by mitochondrial lipid membrane staining, did not differ between treated and non-treated conditions (Figure 3e). Similarly, the nuclear localization of the transcription factor NRF1 and the co-activator PGC1-alpha, two key proteins involved in mitochondrial biogenesis, did not differ between conditions, likely suggesting other mechanisms involved in mitochondrial biogenesis activation (Figure 3f,g).

FTY720-P-treated cells increased mitochondrial nucleoid number, reduced their size, and changed their distribution within AC16 cells (Figure 4a–d). Nucleoids are mtDNA-containing submitochondrial structures of 100 nm diameter that are distributed within the mitochondrial network [56]. Apart from mtDNA, these nucleoids also contain several proteins involved in mitochondrial replication, transcription, and topology, such as TFAM, PolgA, PolgB, Twinkle, and mtSSB [57]. Recent studies reveal that mitochondrial fission machinery and mitochondrial nucleoids are interconnected, which is determinant in the regulation of mitochondrial nucleoid size and distribution [58]. On the other hand, DRP1 contributes to nucleoid distribution in the mitochondrial network, as a loss of function of this protein induces mitochondrial nucleoids’ enlargement and hampers mitochondrial respiration [21,22,46]. Supporting this, DRP1 ablation in cardiac cells promotes nucleoid clustering and mutations. DRP1 has been linked to mitochondrial dysfunction and cardiomyopathy [59]. In this context, our data suggest that FTY720-P may promote mitochondrial biogenesis, mitochondrial nucleoids replication, and a DRP1/TFAM-dependent nucleoid segregation into the mitochondrial network, thereby leading to an activation of mitochondrial function (Figure 1, Figure 2, Figure 3 and Figure 4). 

Beyond changes in mitochondrial proteins and mitochondrial dynamics, our data also revealed that STAT3 phosphorylation at tyrosine 705 and JAK (Janus kinase) 2 at tyrosine 1007/1008 was enhanced upon FTY720-P treatment (Figure 5). Of note, JAK2 phosphorylates STAT3 at tyrosine 705, which in turn upregulates STAT3 gene expression [31,60]. STAT3 promotes the expression of downstream target genes involved in apoptosis prevention, and in the promotion of proliferation and immune cell evasion. Consistently, the activation of the STAT3 pathway is required for the anti-inflammatory, antioxidant, and antiapoptotic effects of FTY720 in white matter (WM) ischemic injury and acetaminophen-induced liver injury model [34,35]. In addition, a growing body of evidence also suggests that STAT3 favorably influences mitochondrial function. For instance, mitochondrial respiration and mitochondrial DNA synthesis are dependent on STAT3 expression in embryonic stem cells and muscle stem cells [32,33]. Moreover, STAT3 is required for OXPHOS protein expression to sustain the redox homeostasis in mouse neuronal progenitor cells [61]. Our data revealed that the positive effect of FTY720-P on mitochondrial respiration and TFAM expression was strongly dependent on STAT3 activation, as this effect was blunted using a STAT3 transcriptional activity inhibitor (Figure 6a–h). Moreover, STAT3-mediated effects were selective to mitochondrial function because ECAR was not affected by FTY720-P in cells (Figure 6i–k).

STAT3 has been recently localized in mitochondria [28], thereby suggesting a role for the phosphorylated form of this transcription factor in OXPHOS activity [28]. In addition, STAT3 has also been localized in the endoplasmic reticulum and at mitochondrial-associated membranes [62,63]. Although further studies are still required to clarify the exact role of subcellular localization of STAT3, recent research indicates that its phosphorylation can be enhanced via S1PR1-JAK2 signaling [64]. Supporting this, S1PR1 overexpression also promotes STAT3 phosphorylation [65]. Furthermore, STAT3 activation prevents the opening of mitochondrial permeability transition pores and thus confers cardiomyocyte protection against subsequent apoptosis [66,67].

In summary, and based on the effect of STAT3 transcriptional inhibitor used in our study, we propose that transcriptional function of STAT3 mediates the favorable FTY720-P-mediated mitochondrial effects. Moreover, it could be suggested that the enhancement of S1PR-JAK2-STAT3 signaling by FTY720-P could mediate a direct signal that modulates the STAT3 pathway and the cellular metabolism.

Here, we provide evidence that FTY720-P promotes mitochondrial respiration and mitochondrial biogenesis in the cardiomyocytes cell line AC16 (Figure 1, Figure 2, Figure 3 and Figure 4). Moreover, STAT3 transcriptional activity sustains the FTY720-P-dependent mitochondrial activation (Figure 5 and Figure 6). Future investigations are required to reveal whether FTY720-P can promote mitochondrial activation and mitochondrial biogenesis in in vivo models. 

## 4. Materials and Methods

### 4.1. Cells and Cell Culture 

The AC16 human ventricular derivates cardiomyocytes cell line was obtained from Sigma-Aldrich, St. Louis, MO, USA, SCC109. Cells were grown in DMEM/F12 supplemented with 2 mM L-glutamine, 15 mM HEPES, 12% FBS, 100 U/mL penicillin/streptomycin, and sodium bicarbonate at 37 °C in a humidified atmosphere of 5% CO_2_/95% O_2_. Unless otherwise indicated, FTY720-P treatment studies were carried out using DMEM/F12 supplemented with 2 mM L-glutamine, 15 mM HEPES, 100 U/mL penicillin/streptomycin, sodium bicarbonate, 1% FBS, and 100 nM FTY720-P. Culture medium supplemented with FTY720-P was changed every 24 h. For the inhibition of STAT3, cells were co-incubated with 25 µM of STAT3 inhibitor S3I-201 and 100 nM FTY720-P for 48 h. Culture medium was changed every 24 h. DMSO was used as vehicle control. Passages 3–7 of AC16 cells were used for this study.

### 4.2. Reagents

DMEM (D5030), oligomycin, carbonyl cyanide 3-chorophenyl hydrazine, CCCP (Sigma # C2759), rotenone, antimycin A, STAT3 inhibitor S3I-201, and Tween 20 were purchased from Sigma-Aldrich. [U-^14^C]-glucose was purchased from PerkinElmer. FTY720-P CAY-10006408 was purchased from Cayman, Germany. Hoechst 33342, Rhodamine Phalloidin, MitoSOX Red (M36008), tetramethylrhodamine, ethyl ester, TMRE (T669), TrypLE™ Express Enzyme (1×), without phenol red (12604021) were purchased from Thermo Scientific, Waltham, MA, USA. The protease inhibitor mixture was Complete-Mini from Roche Basel, Switzerland, (11836153001). Immobilon-FL membranes and the phosphatase inhibitor mixture V were from Merck, Darmstadt, Germany. 

### 4.3. Antibodies

The following antibodies were used: Mouse anti-GAPDH, clone 6C5 #MAB374 from Sigma-Aldrich; mouse anti-TOM20 #sc-17764 (F-10), mouse anti-Lamin A/C #sc-376248 from Santa Cruz Biotechnology, Dallas, TX, USA; rabbit anti-STAT3 Y7051679131S, rabbit anti-JAK2 (Y1007/1008) 07-123, mouse anti-OPA1 #612606 from BD Transduction, San Jose CA, USA; rabbit anti-phospho DRP1 (Ser616) #3455, rabbit anti-TFAM (D5C8) #8076S from Cell Signaling Technology, Danvers, MA, USA, #8076; rabbit anti-PGC1-alpha #NBP1-04676SS from Novus; rabbit Anti-NRF1 #ab175932 from Abcam, Cambridge, UK; anti-DNA mouse monoclonal AC-30-10 #61014PROGEN from Thermo Fisher; goat anti-rabbit IgG (H + L) Alexa Fluor^®^ 488 conjugate and goat anti-mouse IgG (H + L) Alexa Fluor^®^ 568 conjugate from Life Technologies, Carlsbad, CA, USA. 

### 4.4. Western Blotting Assay

AC16 cells (2 × 10^6^ cells/well) were seeded on 1% gelatin-precoated 100 mm plates and maintained in complete culture medium for 12 h. After treatments, cells were homogenized in RIPA (150 mM NaCl, 10 mM Tris (pH 7.2), 0.1% SDS, 1% Triton X-100, 1% deoxycholate, 5 mM EDTA, supplemented with protease inhibitor mixture and phosphatase inhibitor mixture V) and centrifuged at 10,000× *g* for 30 min at 4 °C. Then, 20–40 µg of proteins from total homogenates were resolved in 10% or 15% acrylamide gels for SDS-PAGE and transferred to Immobilon-FL membranes (Millipore, Burlington, MA, USA). Primary antibodies were diluted 1:1000 (*v*:*v*) in TBS with 1% BSA and 0.1% Tween 20. GADPH or Lamin A/C were used at 1:20,000 (*v*:*v*) as a loading control. Secondary antibodies were diluted 1:20,000 in 1% BSA and 0.1% Tween 20. Proteins were detected using the Odyssey Infrared Image System (LI-COR).

### 4.5. Mitochondrial and Nuclear Enriched Fraction

AC16 cells (2 × 10^6^ cells) were seeded on 1% gelatin-precoated 100 mm plates and maintained in complete culture medium for 12 h. After treatments, cells were scraped using a mitochondrial extraction buffer (250 mM Sucrose, 1 mM EGTA, 10 mM Hepes, pH 7.4) supplemented with phosphatase inhibitor mixture V and protease inhibitor cocktail, homogenized by 60–80 strokes with a Glass-Teflon 2 mL Potter Elvehjem homogenizer and centrifuged at 600× *g* for 10 min at 4 °C (nuclear-enriched fraction). The supernatant was collected and centrifuged at 7,000× *g* for 10 min at 4 °C (mitochondrial-enriched fraction). The mitochondrial- and nuclear-enriched fractions were resuspended in RIPA buffer supplemented with phosphatase and protease inhibitor cocktail.

### 4.6. Immunofluorescence

AC16 cells (50,000 cells) were seeded on a 1% gelatin-precoated coverslip and maintained in complete culture medium for 12 h. Cells were treated with 100 nM FTY720-P. Afterwards, the cells were fixed with 4% paraformaldehyde for 10 min, washed in PBS, and then permeabilized with 0.1% Triton/PBS (1×) for 15 min. Non-specific labeling was blocked using 5% BSA. Cells were incubated with anti-DNA (1:100, *v*:*v*) or anti-TOM20 (1:100, *v*:*v*) or STAT3 (T705) (1:100, *v*:*v*) antibodies at room temperature for 2 h, followed by incubation with anti-rabbit Alexa 488 (1:400, *v*:*v*) or anti-mouse Alexa 568 1:400 (*v*:*v*) for 1 h. Nuclei were stained using Hoechst 33342 1:10,000 (*v*:*v*) for 10 min and F-actin was stained with Rhodamine Phalloidin 1:200 (*v*:*v*) for 30 min. Cells were analyzed in an Elyra PS1 + AiryScan Microscope (Zeiss, Oberkochen, Germany). Fast mode AiryScan super-resolution or SIM (super-resolution illumination microscopy) were used to visualize the samples. The images were processed using ImageJ software 1.53t (NIH). 

### 4.7. Mitochondrial Morphology Analysis and Mitochondrial Nucleoid Quantification

Control (untreated) or FTY720-P-treated AC16 cells were stained with anti-TOM20, and z-stack image data were collected using Fast mode AiryScan super-resolution (Elyra PS1 + AiryScan Microscope, Zeiss). For mitochondrial morphology analysis, individual cells were selected. We performed mitochondrial analysis per cell using the ImageJ (NIH) plugin Mitochondrial Analyzer. Block size and C-value for the adaptive threshold were adapted per set of images. This plugin generates mitochondrial shape parameters related to mitochondria number, total volume, and mitochondrial network complexity (mitochondrial branched number, branched endpoint). At least 20–30 cells/sample were counted in triplicate samples per condition and per experiment. For mitochondrial nucleoid analysis, control (untreated) or FTY720-P-treated AC16 cells were stained with anti-DNA and SIM (super-resolution illumination microscopy) images were collected and processed using Elyra PS1 + AiryScan Microscope (Zeiss). Individual cells were selected, and we performed mitochondrial nucleoid analysis using the Analyze particle ImageJ (NIH) plugin. At least 30–50 cells/sample were counted in triplicate samples per condition and per experiment. 

### 4.8. Mitochondrial Oxygen Consumption Rate (OCR)

AC16 cells (25,000 cells/well) were seeded on 1% gelatin-precoated Seahorse Bioscience XF24 plates for 12 h. Cell were treated with 100 nM FTY720-P for 48 h, or co-incubated with 25 µM of STAT3 inhibitor S3I-201 and 100 nM FTY720-P for 48 h. Culture medium was changed every 24 h. Oxygen consumption rate (OCR) was detected. Moreover, 1 µM oligomycin (complex V inhibitor) was used to distinguish the percentage of ATP-linked respiration and the percentage of oxygen consumption required to overcome the mitochondrial proton leak, 1 µM CCCP was used to calculate the maximal mitochondrial respiratory capacity of cells, and 1 µM rotenone (complex I inhibitor) and 1 µM antimycin A (complex III inhibitor) were used to calculate the non-mitochondrial respiration caused by oxidative side reactions. During sensor calibration, cells were kept in a 37 °C incubator without CO_2_ in 500 μL of DMEM 5030 supplemented with 10 mM glucose, 2 mM glutamine, 1 mM pyruvate, and 5 mM Hepes, pH 7.2. Seahorse XFe24 flux analyzer was used to test the mitochondrial bioenergetics analysis. At the end of assay, cells in each plate well were homogenized with Triton 0.1% and the amount of total protein was measured by the bicinchoninic acid (BCA) method. A total of 3–5 independent experiments were performed at least in quadruplicate.

### 4.9. Measurement of Superoxide, Mitochondrial Potential, and Mitochondrial Membrane Staining

AC16 cells (25,000 cells/well) were seeded on 1% gelatin-precoated 24-well plates and maintained in complete culture medium for 12 h. Cells were treated with 100 nM FTY720-P for 48 h. Culture medium was changed every 24 h. To measure mitochondrial superoxide and mitochondrial membranes, cells were loaded with freshly prepared 2.5 µM MitoSOX Red and/or 50 nM Mitotracker Deep Red for 30 min. To measure mitochondrial potential, cells were loaded with 100 nM TMRE for 20 min. As a mitochondrial potential positive control, cells were co-incubated with 20 µM CCCP and the probe for 20 min. Then, cells were washed with PBS (1×), detached by trypsinization using TrypLE™ Express Enzyme (1×), and the reaction was stopped with DMEM 2.5% FBS, without phenol red. A total of 5,000–10,000 cells/sample were analyzed in 3–6 independent experiments performed in triplicate. Analysis was performed using a Spectral Cytek™ Aurora Flow Cytometry System. For data analyses, the mean fluorescence intensity (MFI) was used to calculate the staining of the FTY720-P-treated cells with respect to the untreated control AC16 cells measured in each experiment.

### 4.10. Oxidation of Metabolic Substrates

AC16 cells (150,000 cells/well) were seeded on 1% gelatin-precoated 12-well plates and maintained in complete culture medium for 12 h. Cells were treated with 100 nM FTY720-P for 48 h. On the day of the assay, cells were washed in Krebs-Ringer bicarbonate Hepes (KRBH) buffer (135 mM NaCl, 3.6 mM KCl, 0.5 mM NaH_2_PO_4_, 0.5 mM MgSO_4_, 1.5 mM CaCl_2_, 2 mM NaHCO_3_, and 10 mM Hepes, pH 7.4). To measure glucose oxidation, cells were incubated with 0.1 μCi/mL (U-^14^C) glucose in KRBH buffer for 5 h at 37 °C without CO_2_. The incubations were stopped using 0.5 M H_2_SO_4_. ^14^CO_2_ precipitated in filter papers soaked with KOH 0.1 N o/n at room temperature. Then, these filter papers were added to scintillation tubes for radioactivity measurement.

### 4.11. Expression of Results and Statistical Methods

Data are presented as mean ± SEM of several independent experiments (ranging from 3 to 5). Data were subjected to Student’s *t*-test or one-way ANOVA (for multiple comparisons), and comparisons between groups were performed using a protected Tukey’s *t* test. A *p*-value < 0.05 was chosen as the limit of statistical significance.

## Figures and Tables

**Figure 1 ijms-24-07374-f001:**
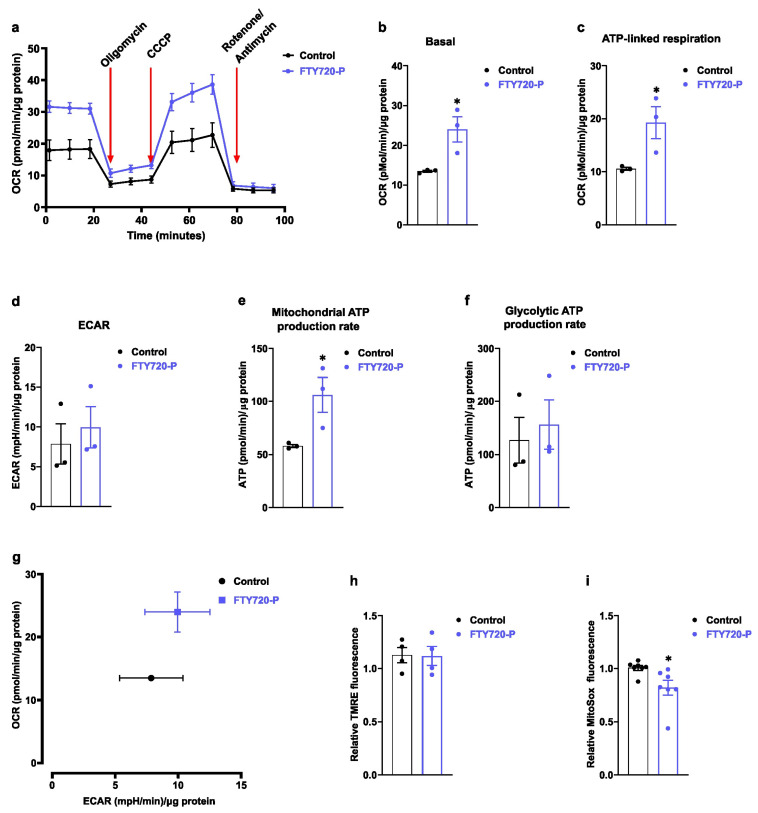
FTY720-P increases mitochondrial function. (**a**) Representative oxygen consumption rate (OCR) profile using Seahorse real-time, live-cell respirometry of AC16 cells treated with FTY720-P for 48 h. Oligomycin (complex V inhibitor) was used to distinguish the ATP-linked mitochondrial respiration and the proton leak across the mitochondrial membrane. CCCP was used to calculate the maximal mitochondrial respiration. Non-mitochondrial OCR was determined by addition of Rotenone and Antimycin A. (**b**–**d**) AC16 cells were treated with FTY720-P for 48 h and then (**b**) basal, (**c**) ATP-linked respiration, and (**d**) extracellular acidification rate (ECAR) were analyzed using Seahorse respirometry. (**e**,**f**) Mitochondrial and glycolytic ATP calculated using OCR and ECAR data obtained by Seahorse respirometry. Data were subjected to unpaired *t*-test. Data are mean ± SEM. * *p*-value ˂ 0.05 vs. control cells. Graphs represent 3 independent experiments performed in quadruplicate. (**g**) Energy profile of AC16 cells determined as a basal OCR in function of basal ECAR in the indicated conditions. (**h**) Mitochondrial potential was detected using the TMRE potentiometric probe. Cells were incubated with FTY720-P for 48 h and then cells were stained with 100 nM TMRE for 20 min. Data are mean ± SEM. Graphs represent 4 independent experiments performed in triplicate. (**i**) Mitochondrial superoxide was detected using MitoSox probe. Cells were incubated with 2.5 μM MitoSox for 30 min, then washed and analyzed by flow cytometry. Data were subjected to unpaired *t*-test. Data are mean ± SEM. * *p*-value ˂ 0.05 vs. control cells. Graphs represent 7 independent experiments performed in triplicate.

**Figure 2 ijms-24-07374-f002:**
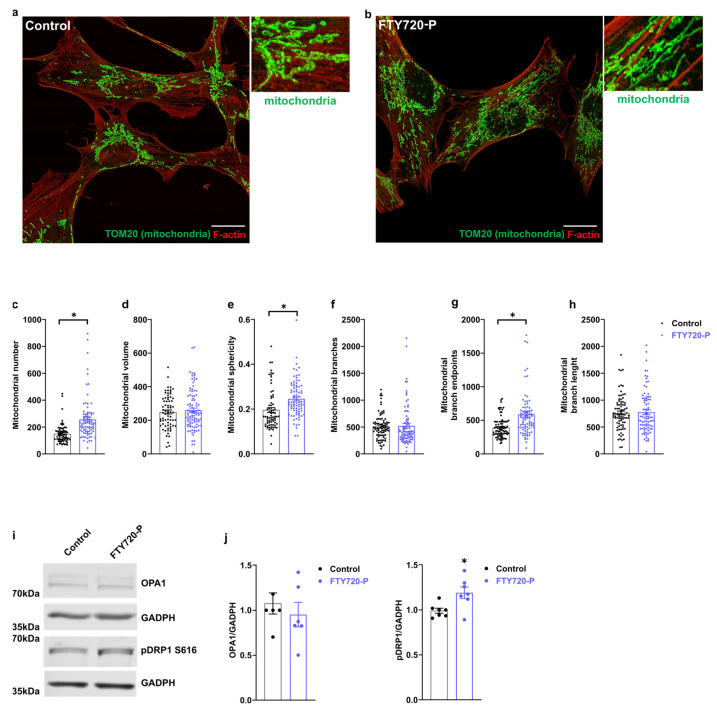
FTY720-P alters mitochondrial morphology. Panels (**a**,**b**) are representative super-resolution images (AiryScan) of mitochondrial morphology in AC16 cells. (**a**) Control AC16 cells incubated with the vehicle, (**b**) AC16 cells incubated with 100 nM FTY720-P for 48 h. The mitochondrial network was stained with anti-TOM20 antibody (green), F-actin was labeled using Rhodamine Phalloidin (red). Scale indicates 20 µm. (**c**) Mitochondrial number, (**d**) mitochondrial volume, (**e**) mitochondrial sphericity, (**f**) mitochondrial branches, (**g**) mitochondrial branches endpoint, (**h**) mitochondrial branch length. At least 20–30 cells/sample were counted in triplicate samples per condition and per experiment. Mitochondrial analysis per cell was performed using ImageJ (NIH) plugin Mitochondrial Analyzer. Data were subjected to unpaired *t*-test. Data are mean ± SEM. * *p*-value ˂ 0.05 vs. control cells. (**i**) OPA1 and pDRP1-S616 protein levels were detected in total protein extract of cells treated with 100 nM FTY720-P for 48 h. (**j**) OPA1 and pDRP1-S616 densitometric quantification. Data were subjected to unpaired *t*-test. Data are mean ± SEM. * *p*-value ˂ 0.05 vs. control cells. Graphs represent 3–6 independent experiments.

**Figure 3 ijms-24-07374-f003:**
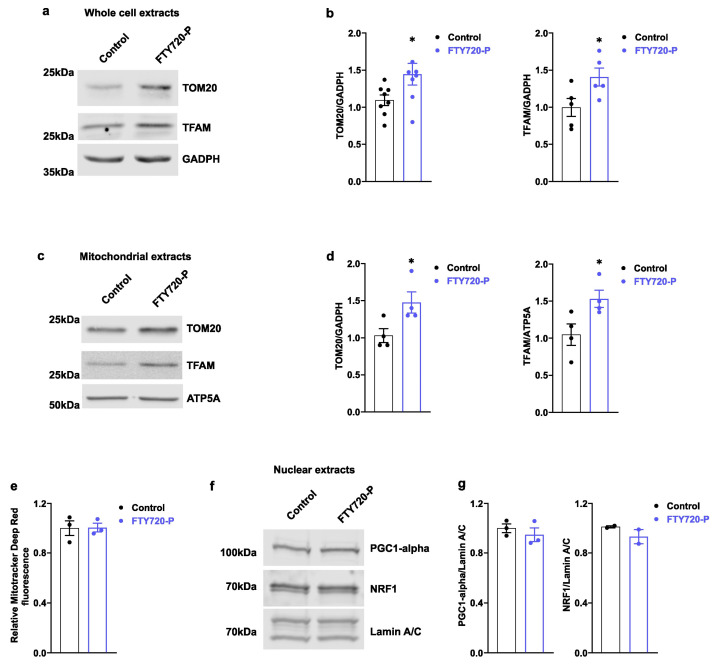
FTY720-P increases TFAM and TOM20 protein expression. AC16 cells were treated with 100 nM FTY720-P for 48 h. (**a**,**b**) Immunodetection of TFAM and TOM20 in total protein extracts of AC16 cells treated with FTY720-P for 48 h. GADPH was used as a loading control. Data were subjected to unpaired *t*-test. Data are mean ± SEM. * *p*-value ˂ 0.05 vs. control cells. Graphs represent 5–7 independent experiments. (**c**,**d**) Immunodetection of TFAM and TOM20 in mitochondria protein extracts of AC16 cells treated with FTY720-P for 48 h. ATP5A (complex V subunit) was used as a loading control. TOM20 and TFAM densitometric quantifications are shown in panel (**d**). Data were subjected to unpaired *t*-test. Data are mean ± SEM. * *p*-value ˂ 0.05 vs. control cells. Graphs represent 3–4 independent experiments. (**e**) Mitochondrial membrane was stained with 100 nM Mitotracker Deep Red for 20 min and analyzed by flow cytometry. Data are mean ± SEM. Graphs represent 3 independent experiments performed in triplicate. (**f**,**g**) Immunodetection of PGC1-alpha and NRF1 in nuclear extracts of AC16 cells treated with FTY720-P for 48 h. (**g**) PGC1-alpha and NRF1 densitometric quantification. Data are mean ± SEM. Graphs represent 2–3 independent experiments.

**Figure 4 ijms-24-07374-f004:**
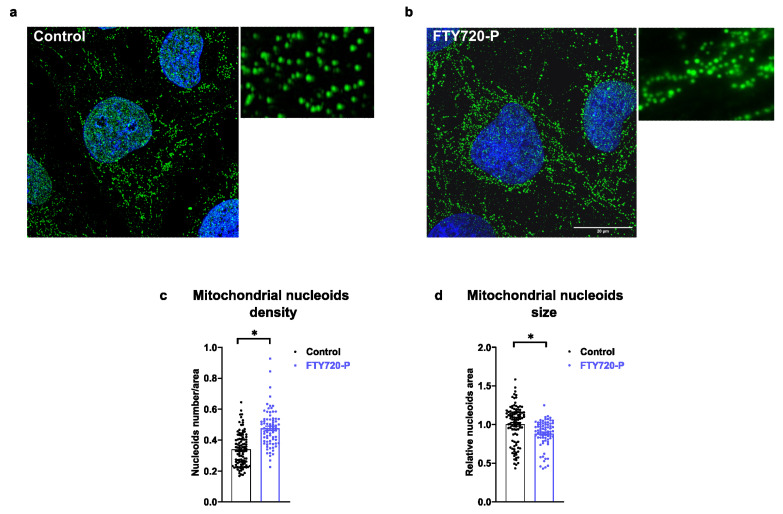
FTY720-P increases the number of mitochondrial nucleoids. AC16 cells treated with 100 nM FTY720-P for 48 h. Panels (**a**,**b**) show mitochondrial nucleoids stained with anti-DNA antibody (green) and visualized with SIM (super-resolution illumination microscopy). (**a**) Control AC16 cells incubated with the vehicle, (**b**) AC16 cells incubated with 100 nM FTY720-P. Nuclei were stained with DAPI (blue). (**c**) Quantification of mitochondrial nucleoid density is represented as mitochondrial nucleoids/cellular area; (**d**) mitochondrial nucleoid size. At least 30–50 cells/sample were counted in triplicate samples per condition and per experiment. Acquired images were processed with ImageJ software (NIH). Data were subjected to unpaired *t*-test. Data are mean ± SEM. * *p*-value ˂ 0.05 vs. control cells. Scale indicates 20 µm.

**Figure 5 ijms-24-07374-f005:**
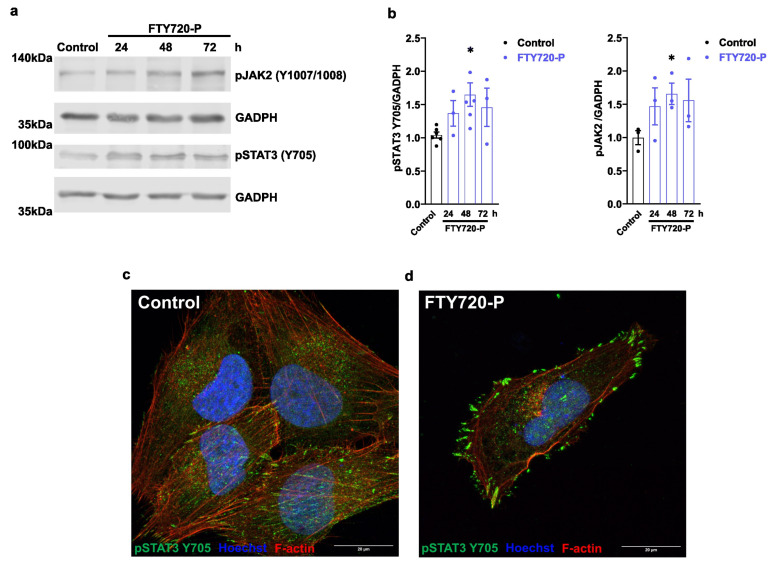
FTY720-P increases JAK2 and STAT3 phosphorylation. (**a**,**b**) AC16 cells treated with 100 nM FTY720-P for 24 h, 48 h, and 72 h. (**a**) JAK2 phosphorylation (Y1007/1008) and pSTAT3 phosphorylation (Y705) were evaluated by Western blot in total protein extract of AC16 cells. GAPDH was used as a loading control. Densitometric quantification is shown in (**b**). Data were subjected to unpaired *t*-test. Data are mean ± SEM. * *p*-value ˂ 0.05 vs. control cells. Graphs represent 3–5 independent experiments. (**c**,**d**) Representative super-resolution images (Airyscan) of cells stained with pSTAT3 antibody Y705. (**c**) Control AC16 cells incubated with the vehicle, (**d**) AC16 cells incubated with 100 nM FTY720-P for 48 h. pSTAT3 Y705 (green), Rhodamine Phalloidin (red), and nuclei were stained with Hoechst 33342 (blue). Scale indicates 20 µm.

**Figure 6 ijms-24-07374-f006:**
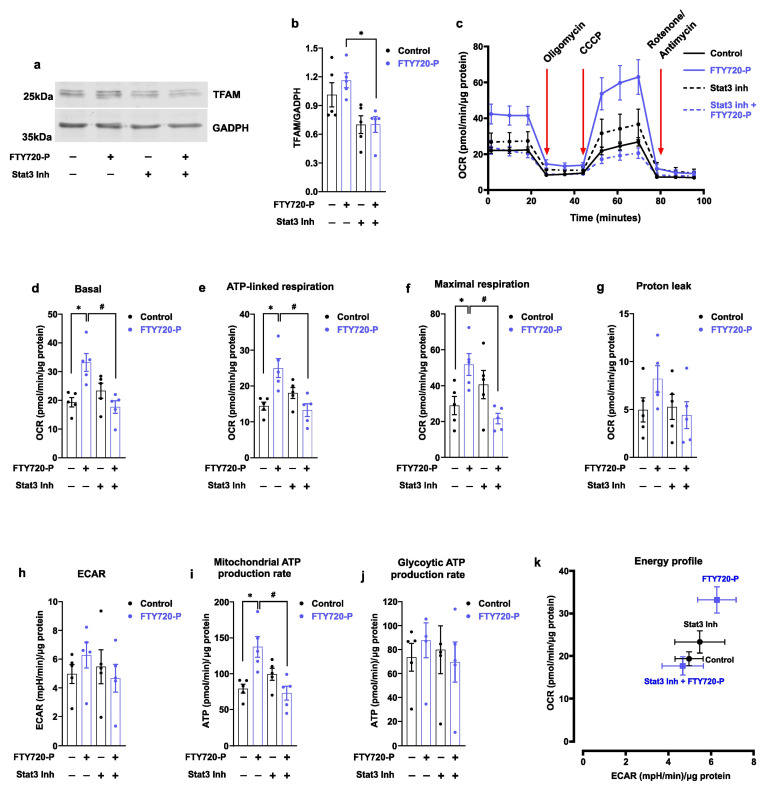
STAT3 inhibitor decreases FTY720-P-dependent mitochondrial activity. AC16 cells were co-treated with 100 nM FTY720-P and/or 25 µM STAT3 inhibitor S3I-201 for 48 h. (**a**) STAT3 inhibitor decreases TFAM protein expression. TFAM was detected in total protein extracts using Western blot. GADPH was used as a loading control. Densitometric quantification is shown in (**b**). Data were subjected to unpaired *t*-test. Data are mean ± SEM. * *p*-value ˂ 0.05 vs. FT720-P-treated cells. Graphs represent 5 independent experiments. (**c**–**k**) Mitochondrial respiration was performed using Seahorse real-time, life cell respirometry. (**c**) Representative oxygen consumption rate (OCR) of AC16 cells co-treated with 100 nM FTY720-P and/or 25 µM STAT3 inhibitor S3I-201 for 48 h. (**d**) Basal, (**e**) ATP-linked respiration, (**f**) maximal respiration, (**g**) mitochondrial leak, and (**h**) extracellular acidification rate (ECAR) were analyzed using Seahorse respirometry. (**i**) Mitochondrial and (**j**) glycolytic ATP calculated using OCR and ECAR data obtained by Seahorse respirometry. (**k**) Energy profile of AC16 cells determined as a basal OCR as a function of basal ECAR in the indicated conditions. Data were subjected to one-way ANOVA. Data are mean ± SEM. * *p*-value ˂ 0.05 vs. control cells; # *p*-value ˂ 0.05 vs. FT720-P-treated cells. Graphs represent 5 independent experiments performed in quadruplicate.

## Data Availability

The data presented in this study are available on request from the corresponding author.

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
