# Peer review of "FTY720-P, a Biased S1PR Ligand, Increases Mitochondrial Function through STAT3 Activation in Cardiac Cells"

_ijms, 2023, doi:10.3390/ijms24087374_

Round 1

Reviewer 1 Report

This article is very comprehensive and is written in a quite deep and constructive manner. 

I have a few suggestions, mainly on grammar corrections (which may lead to sense corrections). 

1) In the Abstract, lane 19: it is better to write “is an FDA-approved” instead of “is a FDA-approved”. 

2) In the lanes 20 and 40, the word “egression” exchange to “egress” (=exit). Moreover, in the lane 293 this term is used correctly (egress).

3) In the lane 61, the word “dynamic” exchange to “dynamics”.

4) In the lane 114, could it be that the word “revealed/showed” is missed in between “respirometer” and “an increase”?

5) Except the fact that this study uses only in vitro models, what could be other possible limitations of the study? 

Reviewer 2 Report

It can be accepted as it is.

Author Response

Response: We thank the referee for taking the time to evaluate our manuscript.

Reviewer 3 Report

Munoz at al. have reported a study in which they investigate the effect of phosphorylated FTY720 (FTY720-P)on mitochondrial function. They have found that this phosphorylated compound when added to cultured cardiac cells, increases mitochondrial respiration rates.  They then explored why this would be and found that it increased STAT3 activation and inhibiting STAT3 also inhibited the effects of FTY720-P on mitochondrial respiration. In addition, FTY720-P increases the number of mitochondrial nucleoids and alters mitochondrial morphology. The main issue I have with this manuscript is how data is presented and the clarity of explaining the results and the experiments.  It is difficult to refute what they are claiming (for example mitochondrial biogenesis) without being able to even understand their argument.  None of the experiments need to be repeated (unless they only did one Seahorse), but clarity is the major issue here. 

Major Comments: 

1.     This manuscript was very confusing to read because both the abstract and the introduction start off by discussing FTY720 then switch almost mid-paragraph to its phosphorylated form which is a so-called “off label” use. They do not explain the significance of phosphorylating the molecule and we do not know whether they are comparing apples to oranges or one type of apple to another.  More explanation is warranted here and a transition sentence making it very clear that they are switching between the non-P and P form of the drug.  For instance on line 53, they finally define FTY720-P which should be the first thing they do before talking about all its effects. 

2.     All Figures:  List the number of repeats in the figure legends. Make bar graphs with dots showing individual measurements.  The authors flip-flop between showing error bars on the control bars and showing no error bars. They should do the analysis the same way every time and show error bars or list in the figure that the error bar is too small to be seen.  This may be helped when individual dots are shown on the graphs.

3.     Figure 1: It is very unclear how many separate Seahorse runs done with separate plates of cells.  The standard is to do at least 3-4 separate plates, not just separate wells within one plate and average those in bar graphs to obtain significance.  Please either list all these details in the methods or in the figure legends being careful to distinguish well number from plate number.  It is fine to show a running line graph of a representative curve (1a).  How were the number of cells in each well normalized? 1H. TMRE is a ratiometric probe.  The ratio of signal outside to inside the mitochondria dictates the membrane potential. It is super unclear what was used in “folds of control”. What is the mysterious control substance?  I couldn’t find anything about TMRE in the methods.

4.     Figure 2: It is not immediately clear that a and b are different.  The differences need to be quantified somehow with an imaging program. C,d Show bar graph of OPA1 as well and instead of saying “folds of control” say what it is divided by (GAPDH in this case).  Needs error bars on first bar.  (Just get rid of the term “folds of control” throughout the paper and actually list what it is divided by)

5.     Figure 3:  If you are going to put a title that says “mitochondrial extracts” make another title for 3a which says “whole cell extracts” or whatever. Then show bar graphs for each western like in Fig 2 and what they are normalized to. 

6.     If the number of mitochondria is the same as shown in Fig 3, then what is shown in the images of Figure 2?  It is not immediately obvious that the mitos in Fig 2 are smaller, just looks like there are less of them.

7.     Figure 5: Error bars on controls, switch the order so Western image corresponds to graphs and put other times on graph. (24, 48, 72h—all need to be shown on bar graph).  C,d Images need to be explained or quantified. The images don’t really look that different from each other.  It is possible the mitochondria in c are just not in the plane of focus.  Maybe a field of cells would make your point better?

8.     How many plates were run?  Is there a representative trace to show like in Fig 1? Are graphs from plates or wells?  How were number of cells normalized?

9.     Discussion needs to mention figure number as you are discussing your data.  This would make the conclusions way more understandable. For instance in line 350, the authors state that FTY720-P may promote mitochondrial biogenesis, but Figure 3 shows there is no change in mitochondrial number. The authors suggest that phospholipid mass does not change, yet still evoke mito biogenesis.  I think without explaining themselves, that this is impossible and points to a lack of biogenesis, but I have no idea which figure they are referring to.

Minor comments:

1.     Line 114 …respirometer show an increase…

2.     Line 119 …in basal and ATP…

3.     Line 130 …bioenergetic parameters…

Round 2

Reviewer 3 Report

Thank you for incorporating my suggestions with the figures. The paper is much more understandable.  However, to make people able to understand this fully, please point out which figures go to which points in the discussion.  All you have to do is put (Fig. x) after the data that applies to that figure.  Otherwise, things are very confusing to people who are not in the field.

Author Response

We kindly thank this reviewer for his/her comments. All of them helped us to improve the manuscript. We have performed the changes in the discussion section suggested by the reviewer.